# Sustainable Energy Autarky and the Evolution of German Bioenergy Villages

**Dariusz Pieńkowski [1,\*]** and **Wojciech Zbaraszewski [2]**

1   Faculty of Economics and Social Sciences, Department of Social Sciences and Pedagogy, Poznań University of Life Sciences, ul. Wojska Polskiego 28, 60-637 Poznań, Poland
2   Faculty of Economics, Department of System Analysis and Marketing, West Pomeranian University of Technology in Szczecin, ul. Janickiego 31, 71-270 Szczecin, Poland; wzbaraszewski@zut.edu.pl
\*   Correspondence: darpie_xl@wp.pl

**Abstract:** The concept of an autarky has a long history and meaning related to its negation and unpopularity. In liberal schools of economics, autarky is usually considered from the perspectives of economic trade protectionism, closed economies, and threats to welfare. Nevertheless, the concept of autarky has gained a new meaning, understood as the local utilization of renewable energy resources from the perspective of their inter- and intragenerational distribution. Local action is shaped by the global perspective. This research consists of three steps. First, a model of energy autarky has been offered based on the system theory. The model shows the variety of the structures and features of energy systems offered in today's debates on energy autarky. Second, the key postulates of sustainable development have been presented to define an autarkical sustainable energy system. Finally, the concept of bioenergy villages in Germany has been presented to illustrate the approach to energy autarky. The research shows that the concept of autarky and single solutions, such as the use of renewable resources, are not themselves a success from the perspective of sustainable development; this misunderstanding is well illustrated by the evolution of the German concept of bioenergy villages into smart villages.

**Keywords:** energy autarky model; energy policy; renewables; economic equilibrium of autarkical systems; absolute and relative autarky

## 1. Introduction

The idea of an autarky has a long history and many facets. There have been fierce economic and political debates on autarky, even in recent history, leading to the negation and unpopularity of this idea. The present policy follows the concept of sustainable development, where it has gained a new meaning as the local utilization of renewable energy resources from the perspective of their inter- and intragenerational distribution; local action is shaped by the global perspective.

In liberal schools of economics (i.e., classical or neoclassical), autarky is usually considered from the perspectives of economic trade protectionism, closed economies, and threats to welfare [1–3]. Moreover, the term was vastly developed by the planners of centralized economies in Germany and socialistic countries [4–6]. These analyses usually pointed out the economic inefficiency and authoritarianism (autarchy) of these economic and social orders. Therefore, the concept of autarky in terms of freedom, paradoxically has a bad reputation in liberal economics.

The remarkable economic doctrine in the light of economic protectionism was mercantilism. The economic view, which was particularly well developed from the sixteenth to the eighteenth century, was that economic power is achievable through gold and silver stocks. An unequal balance of trade was a remedy for both full employment and bullion (gold and silver) accumulation. Various means of

protecting economies became popular, such as custom duties or other legal regulations of economic exchange between countries, usually pictured by the example of the Navigation Acts in England originally adopted in 1651 [7,8].

However, the concept of energy autarky related to the idea of sustainable development has different origins from the liberal economic debate. Today's discussion is often linked with the idea of sustainable development. Since local activity refers to global thinking, energy autarky is driven by transnational policy recommendations and agreements [9–11] that originated from socio-economic and ecological concerns and the global perspective of just human development.

In particular, the Cocoyoc Declaration produced by the members of a symposium organized by the UNEP (the United Nations Environment Programme) and UNCTAD (the United Nations Conference on Trade and Development), Patterns of Resource Use, Environment, and Development Strategies, was eminent in this field [12]. The idea of sustainability was presented in terms of self-reliance as being dissimilar to autarky: "It implies mutual benefits from trade and cooperation and a fairer redistribution of resources satisfying the basic needs. It does mean self-confidence, reliance primarily on one's own resources, human and natural, and the capacity for autonomous goal-setting and decision-making. [ . . . I]t implies decentralization of the world economy [ . . . , and] it also implies increased international co-operation for collective self-reliance" [13] (pp. 897−898). Future documents on sustainable development policy, for example Agenda 21, utilized the terms "self-sufficiency" and "self-reliance" interchangeably [14] (for example, Articles 32.5.d, 3.8.l, 14.16, 14.26, or 14.93), but it is a rather semantic interpolation to separate this view from the negative connotations and associations of the term "autarky" in political and social sciences. Self-reliance is perceived as an element of the global climate mitigation policy and the way of allocating social, economic, and natural resources from the inter- and intragenerational perspectives.

Müller et al. [15], referring to the original definition of sustainable development presented in the Brundtland Report in 1987, defined a sustainable energy system as follows: "it must be capable of providing the energy services demanded by the current population, whilst ensuring that future generations find the economic, social and ecological resources they require" [15] (p. 5800). This postulates the criteria for the evaluation of the final outcomes of the action that should be undertaken toward achieving energy sustainability. The practical implications are the ongoing processes of transformation and the examination of different solutions. The ideas of self-reliance and self-sufficiency presented in this document were included in the action plan for the upcoming summit.

Agenda 21, as a plan of action for the twenty-first century, was implemented during the UN World Summit in 1992. This detailed document of action widely utilized the ideas of self-reliance and self-sufficiency. The self-sufficiency strategy, as a sustainable method of energy production, was articulated in Chapters 14 (Article 14.16 or 14.93) and 32 (Article 32.5), respectively, on promoting sustainable agriculture and regional development and strengthening the role of farmers [14]; these were mostly related to farming and rural areas. The postulates of energy efficiency, universal access to energy, and the role of renewable energy resources in sustainable development were also articulated in Chapter 4 on changing consumption patterns [14].

The UN World Summit generally reaffirmed the above postulates [16] and reformulated them in a new document named "The Future We Want" [17]. Finally, the 2030 Agenda for Sustainable Development by the United Nations from 2015 presented them in the form of 17 Millennium Development Goals and 169 targets [18]. It emphasized that the goals are "integrated, indivisible and balance the three dimensions of sustainable development: the economic, social and environmental" [18] (p. 1).

The energy issues were postulated within the seventh goal, entitled "Ensure access to affordable, reliable, sustainable and modern energy for all" [18]. This goal was further developed into the three following strategies: (1) universal access to modern energy services, (2) the improvement of energy efficiency, and (3) the increased use of renewable sources [19]. The socio-political postulates predominate in the first strategy, which refers to inter- and intergenerational just energy distribution. The next one mirrors the economic dimension, while the third is mostly concerned with environmental

issues. According to the concept of sustainable development, an energy village has to implement all these strategies simultaneously.

The new documents published since 2012 did not use the terms "self-sufficiency" or "self-reliance" (or "autarky") directly, although "The Future We Want" refers to the guidelines of the Food and Agriculture Organization of the United Nations in Article 168 on the institutional framework of sustainable development. The latter coined the concept of energy use in terms of "energy-smart" farming linked to the model of integrated food–energy systems (IFESs) [20], "in which food and energy are produced concomitantly on farms to achieve sustainable crop intensification" [20] (p. 35). The concept is mostly related to bioenergy, that is, energy from biomass generated from agriculture, and assumes only varied degrees of energy self-sufficiency on small farms. Nevertheless, the energy self-sufficiency in the context of all renewable energy sources is seen as a sustainable energy-smart strategy for local communities or small farms; it also emphasizes additional revenues from the potential energy excesses if the on-site demand is met [21].

The concept of sustainable energy and self-sufficiency is thus usually related to these priorities provided by the sustainable development debate. For example, Lund [22] pointed out three main strategies of sustainable energy development related to the following technologies: (1) energy savings (consumption), (2) energy efficiency (production), and (3) transition to renewable energy sources. This technological angle is consistent with the above postulates, although the question of universal access has been tackled less. Dincer [23] also included the political problems of the growing demand for energy that followed the population growth.

Deutschle et al. [24] refined the main strategies into an open-ended list of the goals of energy sustainable systems, including autarky, such as energy, resource, and cost efficiency; climate change neutrality; and renewable energy sources. A sustainable energy village should be evaluated with all these criteria and the degree to which it meets the requirements defined by its goals. Therefore, an energy autarkical village with efficient technologies based on fossil energy sources can be less sustainable than a village that is dependent on the solar energy provided by a neighboring village. Similarly, an autarkical village with efficient energy grid services, which invests in a costly, expensive to maintain, and resource-intensive energy storage infrastructure, fails to satisfy the sustainability criteria in terms of resource and cost efficiency. In turn, a combination of such sustainable energy indicators for local initiatives was investigated by Neves and Leal [25]. They defined 14 state and 4 policy indicators related to social, economic, and environmental dimensions. The study included criteria such as household energy intensity, the renewable share of energy, and the ratio of energy green jobs to the population, and awareness-raising campaigns and locally available financial schemes among the political ones. The latter reflect political and social acceptance and participation, which are also crucial elements of sustainable development.

The research in this paper investigated two hypotheses:

**Hypothesis 1.** *Sustainable energy autarky is not a concept of social and economic isolation or political power, such as in the political and economic debates on economic protectionism.*

**Hypothesis 2.** *Energy autarky itself does not have to meet the postulates of sustainable development.*

Sustainable energy autarky is not a concept of isolation but an idea of using global resources locally. In other words, unlike in the liberal economic debate, the circles of energy are locally locked (also in the broader sense, as in the case of relative autarky) to secure resources for a local community in terms of the postulates of sustainable development, that is, to secure them for other societies and generations. However, energy autarky itself does not have to meet the postulates of sustainable development, nor single solutions such as the use of renewable resources.

## 2. Methodology

The problems with the understanding of autarky have originated from the many facets of the term and its different implications since the very beginning. This is also the case with energy autarky and its implications in terms of sustainable development. Therefore, a model of energy self-sufficiency is needed (the term is used interchangeably with autarky in this paper) that captures the basic characteristics of energy autarky. This will then help to put in place the sustainable conditions for energy autarkical systems.

The model offered in this paper is based on the debates and research on energy autarky presented in literature. For example, Engelken et al. [26] emphasized the differences between relative and absolute (physical) energy self-sufficiency. Both systems assume that 100 per cent of energy demand is satisfied from local resources. The former, also named autarky "on balance" [24,27], on grid [27,28], or grid-connected [27–29], is defined as the ratio between the locally created energy supply and the local energy demand. The energy is fed into an energy grid (e.g., the electricity grid) and provided for consumers on demand. Energy production and consumption are asynchronous and separated; thus, there are both supply surpluses and supply shortages. In turn, absolute self-sufficiency (or off-grid, grid-alone, strong, or physical) assumes a lack of external energy recipients and suppliers [24,26–28]. The time lag between consumption and production is limited by the local energy storage infrastructure, and the processes are more synchronized and balanced "on the spot."

Moreover, energy production can be analyzed at different levels, such as those spatial levels presented by Engelken et al. [26]: households, organizations (e.g., firms), municipalities, or states; it can be further completed with regions (e.g., provinces), groups of states (e.g., the European Union), or the global level [30]. Additionally, a variety of energy carrier types (e.g., electricity or heat) has been mentioned, which can be defined as functional self-sufficiency [24,26]. Consequently, a local electricity system can be self-sufficient while a heating system is not. Moreover, the supply and demand for energy are asynchronous, and relative self-sufficiency in particular needs a time horizon [24,30]. In practice, implicit assumptions are usually made about the calendar year balance, although from a longer perspective, self-sufficiency can be achieved to a much lower degree in the many years included in the wider time analyses (e.g., because of very cold weather in some years). Finally, Engelken et al. [26] assumed that a continuous degree of energy self-sufficiency can be observed in terms of the relationships between energy supply and energy demand [26]. In turn, McKenna et al. [28] proposed a special category for the transitional stages, such as "targeted at autarky," with a contractual threshold value (they accounted for 50 per cent of self-sufficiency). In this paper, this threshold was accepted because it assumes that energy consumed from local resources exceeds those delivered by energy grids; moreover, a similar approach has been presented in the debates on bioenergy villages in Germany.

The research consisted of three steps. First, a model of energy autarky has been offered based on the system theory originally described in science by Bertalanffy [31]. A biological system is an entity that consists of functional interrelated elements demarcated from their environment. The structure of the system's elements is determined by its goals. The model shows the variety of the structures and features of energy systems offered in today's debates on energy autarky according to the different environmental conditions. Second, the key postulates of sustainable development have been applied to define an autarkical sustainable energy system. The postulates were mostly based on the aforementioned UN (United Nations) documents and research on sustainable energy development. Finally, the evolution of the concept of bioenergy villages in Germany has been presented, in the context of the sustainable energy autarky model, to illustrate how the narrow approach to energy autarky resulted in the extensive production of biomass for power installations. This example illustrated the problem of narrow approaches to sustainability from the perspective of autarkical energy systems. The problems related with the first concepts of bioenergy villages challenged the sustainability postulates. Therefore, the idea in Germany has evolved into the new concept of smart or resilient villages and autarky at higher spatial levels.

## 3. Results

The approaches to autarky presented in this paper consist of many assumptions that should be clarified and interrelated in a consistent model of autarky. The model offered in this paper organizes and clarifies the key parameters that should be estimated when energy autarky systems are analyzed. It contains the key parameters that are usually only partly analyzed in research thus far. Moreover, it allows one to understand the relationships between an autarkical energy system and sustainability.

### 3.1. The Model of Energy Self-Sufficiency

The key characteristics of energy self-sufficiency are plotted in Figure 1. The *X*-axis presents the relationship between the supply of local energy sources and the local demand for energy in a system, while the *Y*-axis reflects the strategies to deal with surpluses. The axes cross at equilibrium (i.e., the ratio of supply to demand equals 1) on the *X*-axis and the point of a lack of grid feeding on the *Y*-axis. The values of the *X*-axis range between zero and infinity. The *Y*-axis represents two extremes, as follows: the local storage capacities and the degree of grid feeding.

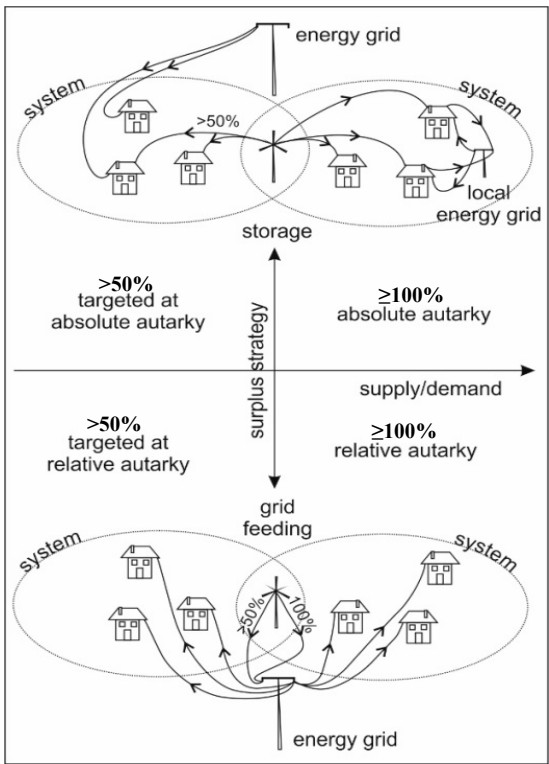

**Figure 1.** The model of energy self-sufficiency. Source: own elaboration.

The model is founded on the following assumptions discussed in the literature on energy autarky:

- a system can be analyzed at different levels of its functional (e.g., in terms of energy carriers or economic sectors) and spatial characteristics;
- energy is a means for the socio-economic functions of a system, and the criteria for its boundaries are defined by the functions;
- all energy sources (i.e., nuclear energy carriers, fossil fuels, and renewable energy resources) are included;
- an energy grid is an energy infrastructure in the environment of a system with outside management of the system;

- a local energy grid is an energy infrastructure fed only by local energy producers, and it is considered a local storage system (the potential exchange between the parts of the system or storage in a locally provided infrastructure);
- an energy grid has unlimited capacity for energy feeding;
- local energy storage capacity occurs, although it is temporarily limited;
- the storage of energy in a relative autarkical system is marginal and economically unreasonable;
- the surplus beyond the storage capacity of an absolute autarkical system is marginal (it will generate a demand or not be produced);
- there is a contractual assumption about the time horizon of relative autarky (usually one year);
- a system is autarkical when the ratio between the locally generated supply and the local demand is greater than or equal to 1;
- the degree of autarky starts from zero (i.e., a continuous scale);
- the system targeted at autarky is defined by a contractual assumption about the minimum ratio between the locally generated supply and the local demand;
- an autarkical system is possible with the degree of each autarky (relative and absolute) above zero and with the mixed strategy of dealing with surpluses (i.e., certain degrees of both grid feeding and storage).

There are some issues in the model that need further explanation. The crucial point to understanding a system analysis is the spatial or functional boundaries delimiting a system and its environment. Energy in social systems is a means of developing its socio-economic functions. Therefore, the system boundaries refer to these functions [15]. Müller et al. [15] suggested administrative borders, such as municipal borders; they can be developed further to provinces, regions, countries, and so on. It is also possible to use a typical economic entity, such as a household or an organization, and many other criteria.

However, the idea is that the criteria should be applied consistently and take into account the socio-economic functions of the system. For instance, if a municipal entity is analyzed, the degree of autarky refers to all the firms and households, as well as the communal infrastructure within the administrative borders of the municipality. Consequently, in an autarkical municipality, the entire energy demand (i.e., electricity and heating) of all the entities is met by its resources, or it may only be an autarkical commune in terms of heating. For example, most German bioenergy villages do not meet this criterion and should be considered to have only a certain degree of autarky; that is, they are "targeted at autarky." They exceed the threshold of 50 per cent of local energy production and the participation of all the members is usually incomplete; the "way towards a bioenergy village"—used in the terminology of the German founders of the concept—means exceeding the threshold of energy production, and (mostly) relative autarky goals are postulated [32,33] (the criteria are also evaluated within the same organization, and there are many definitions of the concept; see Heck et al. [33] and Ruppert et al. [34].)

Furthermore, relative autarky and absolute autarky are delimited by the criterion of surplus strategies. The criterion assumes an energy storage infrastructure only in the systems with absolute autarky. Compared with grid services, its high cost, lower efficiency, and other disadvantages, such as an unpredictable supply of many renewable energy sources, discourage investment. Kaundinya et al. [29] and Rae and Bradley [35] emphasized the unlimited storage capacity of on-grid systems and the separation between the local energy supply and the local energy demand, which makes it difficult to balance the two. The latter can threaten the sustainability postulates discussed in the next section; therefore, it will be discussed further.

The local energy grid or stand-alone mini-grid [21] solutions are treated as an element of energy storage, and they usually form a typical energy storage infrastructure, such as batteries or seasonal heat pit storage (for more details see References [36,37]); however, they may be different forms of the exchange of surpluses between local individual producers and local consumers. Moreover, it is clear

that all the systems aimed at autarky are mixed autarkical systems. Some of the systems can never reach the goal, and others can develop technology with huge energy storage capabilities, for example, the solar system in Dronninglund in Denmark with 60,000 m$^3$ of seasonal pit heat storage (the capacity of the storage is 5100 MWh) [38].

### 3.2. Energy Autarky and the Concept of Sustainable Development

Autarky, as discussed in the previous sections, can be investigated from different perspectives, and the idea is not exclusively linked to energy resources or the concept of sustainable development. An autarkical system can be unsustainable or part of autarchic ideas resulting in isolation, conflicts or ecological burdens. Therefore, energy autarky as part of a sustainable development strategy has to be understood in the context of its postulates. The concept of sustainable energy autarky should satisfy the key postulates of sustainable development, because by itself, it does not meet all these criteria.

Therefore, the debate about relative and absolute autarky refers to the economic mechanism of energy distribution. The investments in the new models of energy production are usually heavily financed from external funds, and a mechanism is usually worked out to support the energy supply mboxciteB33-sustainability-573120,B39-sustainability-573120. Absolute autarky limits the balance between production and consumption and the locally owned energy storage infrastructure. The produced energy is consumed on the spot or stored as a reserve for periods of reduced supply. The supply is thus limited by the technology and locally accessible resources. However, in the case of relative self-sufficiency, the drivers for achieving equilibrium are determined by the energy exchange conditions between a local system and an energy grid. Therefore, the rules of exchange have to be designed to stop the increase in energy demand followed by the use of more available energy, which illustrates Jevon's effect [40].

Figure 2 presents energy self-sufficiency from the perspective of the economic theory of equilibrium and renewable energy resources. The renewable energy supply and demand are presented in relation to the equilibrium price. The demand and supply are determined by the socio-economic factors, such as income and consumer preferences or production costs and technology. The maximum energy supply (produced and stored if not consumed on the spot) in an autarky is marked by $Q_{max}$. The value is critical for a self-sufficient system in terms of physical conditions (absolute autarky). The supply curve will rise vertically ($S_a$), showing the supply limit of a system at this point (i.e., the price no longer affects the supply). The short-run shortages and surpluses are balanced, and the system is driven by the equilibrium price.

In the case of relative self-sufficiency, the energy exchange conditions are in the form of direct or/and market regulations; the former are particularly common in many developed countries. The direct regulation instruments that oblige grid recipients to receive all the energy production at a fixed price are depicted by the horizontal supply line ($S_r$) at the level of this price. In this extreme case, the consumption and production are determined by the fixed price level. The supply of energy from renewables is provided at the level of the price and it can lead to its extensive consumption (as observed by W. S. Jevons). Self-sufficiency based on grid systems is a more comfortable and secure solution. However, a mechanism is needed that effectively balances the local production and consumption of energy; otherwise, a system will be autarkical but unsustainable (e.g., in terms of resource and economic efficiency).

The problem should already be monitored. For example, on Sunday 23 August 2015, the renewable energy supply in Germany met over 83 per cent of the total energy demand, leading to a fivefold fall in the energy price per MWh on that day [41]; there were 97 h of negative prices in 2013 on the electricity market that resulted from the surplus of energy from renewable sources [42]. The wholesale energy prices in this country with very high energy consumption per capita [40] decreased in 2015 by about 10 per cent per kWh [43,44], and, for example, the costs of the solar technology infrastructure decreased by 30 per cent in 2015–2016 [44]. Kopp et al. [45] predict that the prices on the renewable energy market

will decrease until 2050 by more than double compared with 2010; this can be conducive to the increase in energy consumption.

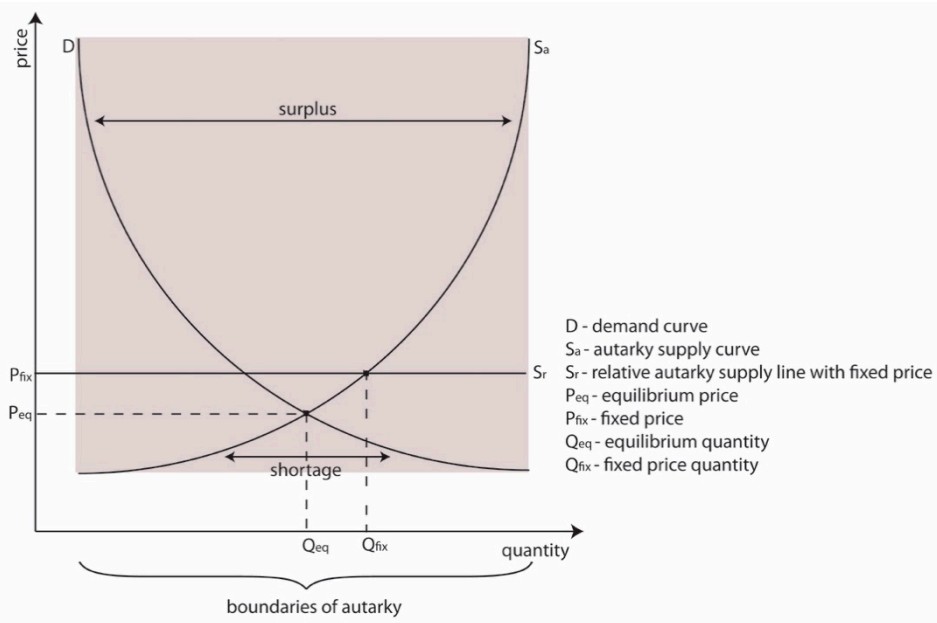

**Figure 2.** An economic model of energy self-sufficiency. Source: own elaboration.

Finally, the issue of renewable resources, which is inextricably linked with the concept of sustainable development in terms of energy resources, should be discussed. Energy sustainability also challenges renewable resources. It may be misleading to assume that the technology related to renewable resources itself is sustainable. For example, Omer [46], in his concept of absolute and relative energy sustainability (of the electricity supply), defined the former as follows: "no depletion of world resources and no ongoing accumulation of residues" [46] (p. 2277). In turn, the degree of sustainability, that is, relative sustainability, was used to compare different energy technologies. Therefore, the author assumed that all renewables are absolutely sustainable.

However, the inefficient production of energy from renewable resources can result in further environmental harm. The renewables represent a variety of technologies with different levels of use efficiency (e.g., see Mani and Dhingra [47]), and extensive use of the resources leads to serious environmental threats. These were widely analyzed by Hadian and Madani [48] using their idea of the relative aggregate footprint (RAF) and some other indicators. The study widely assumed the use of renewables from the economic perspective, as well as the water, climate, and land impacts. For example, the RAF (which ranges between 0 and 100) of the popular biomass crops of *Miscanthus gigantheus* (e.g., see References [49,50]) accounted for a larger footprint (84) than natural gas (60) or coal (about 73), and nuclear energy accounted for the least at about 24. The poor rating of biomass was mostly determined by large water and land footprints. It was seen by the authors as a renewable but not green (i.e., sustainable) energy resource. This resulted from the economic and broader environmental perspective of its evaluation in line with that postulated in the concept of sustainable development.

### 3.3. The Evolution of German Energy Biovillages toward Sustainability

The German energy policy, particularly intensively developed since the 1990s, has aimed at the concepts of local, regional, and national autarky related to renewable resources. The transformation began in 1991 with the coming into force of the Electricity Feed-In Act (*Stromeinspeisungsgesetz*), which guarantees the receipt of electricity energy from the producers of green energy. However, two significant breakthroughs occurred when the Renewable Energy Act (*Erneuerbare-Energien-Gesetz*—EEG) in 2000 and the energy transition policy packet from 2010 (*Energiewende*) came into effect [51–53].

These German energy reforms were primarily linked to the growing debate on climate change resulting from disenchantment with both coal and nuclear energy. The former was directly related to the ecological burdens that were particularly noticeable in the devastated coal-mining areas of eastern Germany after unification in 1990. In turn, nuclear catastrophes, such as those in Chernobyl and Fukushima, strengthened the sense of threat and decisive action toward renewable energy [51]. The energy policy packet intended to close nuclear plants by 2022 and to increase the share of annual renewable power consumption in gross inland consumption by up to 80 per cent by 2050 [53].

These reforms facilitated new models of energy production in line with the key postulates of the concept of sustainable development. They contributed to the idea of energy self-sufficiency and bioenergy villages, which became part of the strategy of sustainable regional development [15] or, more generally speaking, the sustainable energy transition [26]. The process of transformation proceeded with long-term social negotiations supported by environmental impact studies and an economic costs and benefits analysis. External financial support was usually delivered to increase the profitability of these investments. Nevertheless, numerous social and economic problems remain unresolved, despite the fact that new villages are still emerging following the pioneering ideas.

The term "bioenergy village" has evolved since the first village was created in 2005, although it is firmly rooted in three ideas:

- energy autarky;
- bioenergy (i.e., energy produced from biomass);
- rural areas and agriculture.

The first criterion assumes that biomass has to fully cover the local demand for electricity and a minimum of 50 per cent of the local demand for heating; additionally, over 50 per cent of the heating infrastructure should be owned by the local consumers of heat energy and local farmers [34]. In turn, the definition by the FNR (*Fachagentur Nachwachsende Rohstoffe e. V*—Agency for Renewable Resources, financed by the German Federal Ministry of Food and Agriculture) due to the share of renewables accepts a minimum of 50 per cent of the demand for energy being covered by the regional bioenergy supply [32].

However, the updated criteria weaken the share and importance of biomass in energy production, assuming, as was done previously, autarky in electricity and the coverage of a minimum of 75 per cent of local heat energy, both entirely from renewables; additionally, the focus is on raising the efficiency of energy use and production [33]. The transformation has been described semantically by a prefix in brackets: (bio)energy villages [33,34]. Finally, the concept of bioenergy villages has been transformed into a new idea of smart villages, which widely includes the following sustainability factors: (1) electricity, (2) heating, (3) land protection, (4) efficiency, (5) innovation, and (6) social participation [54–56]. The transition was mostly forced to occur by the problems resulting from the biomass footprint [48,57–59].

Moreover, the term "village" (*dorf*) can be misleading if analyzed in terms of the administrative borders of a municipality, as, for example, was recommended by Müller et al. [15]. The German idea refers to an unincorporated community in rural areas; therefore, it can mean an abbey (e.g., Abtei Münsterschwarzach in Schwarzach am Main) or a district (e.g., Altenmellrich in Anröchte), as well as a municipality. Consistent with the system analyses, however, the units lack a common denominator for a statistical comparison at the political level. Additionally, the energy autarky transition is an ongoing process, and the data provided for the analysis are constantly changing and inconsistently provided [32]. The FNR database consists of 139 notifications provided by the authorities of the bioenergy villages in June 2017 and over 45 villages aiming to become bioenergy villages [32]. A notorious report by trend:research [60] (a German commercial research organisation) estimated that there will be over 400 energy autarkical entities in 2020 in Germany. Based on these data, the total population in the energy autarky entities is estimated to be less than 1% of the German population.

The local and regional energy autarky in Germany is still a vital and ongoing process of energy transition. The first successful examples have been achieved, although only in terms of the electricity and heating demand for the building infrastructure (i.e., excluding transport). Despite the evolution of the concept of local energy autarky since the first bioenergy villages were established, the research announced by the German Environment Agency (*Umweltbundesamt*) on the electricity autarky plans by 2050 admits that local electricity autarky is an unrealistic concept, particularly when transport, local commerce, and industry are included [61]. The concept can become even more difficult to realize as soon as the government stops subsidizing the investments.

For example, a very famous energy (absolute) autarky village, Feldheim in Germany, inhabited by 135 people in 31 households, produces energy from numerous renewable resources with 55 wind turbines, 9844 solar modules, a woodchip heating plant, and biogas plants; additionally, electricity storage infrastructure has been built [62]. The cost of the electricity and heating infrastructure (excluding the storage infrastructure) amounted to €2,175,000 [63]; the storage infrastructure additionally cost €12,800,000 [64]. The cost of the investments was thus €110,000 per inhabitant or €483,000 per household. This challenges both the financial ability of the local communities (external support was provided) and the economic profitability of the project; additionally, environmental problems have been observed [65].

## 4. Discussion

The model of autarky illustrates the relationships between the different systems of autarky (i.e., relative and absolute) in terms of both economic (i.e., demand and supply) and technical (i.e., storage strategy) dimensions. It was created to clarify the differences between particular autarkical systems and it emphasizes the key parameters that have to be estimated in the debate on autarkical energy systems (e.g., a time horizon or energy carriers). However, the model clearly shows that autarkical systems satisfy only one sustainability parameter discussed and presented in the documents on the concept of sustainable development, which is energy self-sufficiency. The sustainable path toward the development of energy systems imposes many other locally/regionally specific criteria, which together should satisfy the sustainability goals postulated in the political and scientific debates on economic, ecological, and social dimensions.

For example, the idea of energy self-sufficiency widely presented in the local/regional initiatives of bioenergy villages developed in many European countries [63,66] has evolved since the first villages were established due to the problems generated by narrow approaches to energy autarky. The most advanced and interesting example is found in Germany. The German idea of bioenergy villages shows that the concept of sustainable energy autarky is not a concept of social and economic isolation or political power, such as in the political and economic debates on economic protectionism. The national policy and local or regional initiatives try to meet the postulates presented in the debate on sustainable development, balancing socio-economic and ecological determinants. The complexity of the global perspective challenges many bottom-up, right-oriented initiatives, which are monitored, supported, and channeled into the sustainable paths by governmental organizations.

The German example also shows that energy autarky itself does not have to meet the postulates of sustainable development. A sustainable energy model of transition based on energy decentralization should consider the specific local or regional socio-economic and ecological conditions (e.g., for an absolute or relative autarky), as well as many other factors (e.g., efficiency, social participation, or land protection) as presented in the previous sections.

The local energy autarky concept has to be understood from the perspective of global goals. The costs of local absolute autarky in developed countries and the potential environmental risks caused by the technologies, such as the extensive production of biomass for power installations resulting in monocultures of energy crops, soil sterilization, and severe water shortages, challenge the sustainability of the idea. As a consequence, an insufficiency of local renewable resources does not denote a lack of a sustainability strategy because the local community can be part of a self-sufficient system at the higher level (e.g., a regional level). Energy from renewables is provided from regional resources. Therefore,

the idea of bioenergy villages has evolved into the new concept of smart villages and autarky at higher spatial levels. The current position of the German policy assumes that it is mostly applicable at the higher levels (i.e., regional or national). The concept of local energy autarky is only recommended for some small and specific communities [61].

## 5. Conclusions

The concept of autarky (self-sufficiency and self-reliance) has returned to the economic debate with a new face of sustainable energy production and consumption. However, it is misleading to assume that energy autarky, as well as renewable energy resources themselves, are sustainable. There are additional criteria that should be met to achieve the goals postulated by the concept of sustainable development, such as efficiency or just distribution.

The model of autarkical systems should be used to estimate and clarify one of the parameters of sustainability (i.e., energy self-sufficiency) and the specific strategy of its development due to local/regional conditions. Therefore, all the sustainable postulates discussed in the previous sections should also be taken into account (i.e., social, economic, and environmental ones) when the parameters of an autarkical system are designed. There is no universally sustainable autarkical energy system due to the variable nature of sustainability determinants, such as annual variations of the energy supply from renewables, different conditions for their use, and diverse institutional and cultural patterns of consumption.

The evolution of German bioenergy villages illustrates the search for a sustainable path to developing energy systems. The German experience shows that the bottom-up initiatives should be monitored and supported due to their impact on higher organizational levels, including national and global ones, because the present idea of energy autarky has to meet sustainability goals.

In fact, particularly in developed countries, the concept of self-sufficiency is a relative one. The energy market is regulated, and many direct regulations are provided to support the exchange between producers and consumers; additionally, the grid infrastructure is well developed. Absolute autarky is not even desirable in these countries [30].

However, the system of absolute self-sufficiency is not an abstract concept. This idea is particularly exploited on two extreme levels of analysis: a very low level, such as households, and the highest level, the global ecosystem. In other words, single households and communes separated from the grid infrastructure can achieve absolute self-sufficiency. This is usually the case for many developing countries or spatially separated communities. The second extreme was described well from this perspective in the works by Boulding [67]. Earth was presented as a spaceship with a limited circle of matter and energy; thus, the world was seen as an absolute self-sufficient system. The concept is also reflected in the postulates of a circular economy [68].

Research is needed on existing autarkical energy communities from the perspective of the economic, social, and ecological conditions of their development. Energy autarky (particularly with highly subsidized investments) is usually assumed to be a successful path to sustainability. However, the particular parameters of sustainability of the existing communities should also be investigated in the context of parameters such as the impact of fixed price levels in the relative autarkical systems on the use of renewables, behavioral change in energy use (e.g., in the context of the Jevons effect), or the ecological impact of new socio-technical structures. Finally, the role of energy self-sufficient communities should be researched from the perspective of building a low carbon society and a circular economy.

**Author Contributions:** D.P.: conceptualization, methodology, investigation, writing—original draft preparation. W.Z.: conceptualization, methodology, investigation, writing—review and editing.

**Funding:** This research received no external funding.

**Conflicts of Interest:** The authors declare no conflict of interest. The funders had no role in the design of the study; in the collection, analyses, or interpretation of data; in the writing of the manuscript, nor in the decision to publish the results.

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
