# Peer review of "Sustainable Energy Autarky and the Evolution of German Bioenergy Villages"

_sustainability, doi:10.3390/su11184996_

Round 1

Reviewer 1 Report

The conventional idea of autarky that authors present in association with the theme of sustainable energy is relevant and adequate to explore new conceptual approaches toward the promotion of a low carbon society.

Moreover, the comparison between energy autarky and energy self-sufficiency is coherent and original within the context of circular energy systems based on renewable energy resources at a local scale. Indeed, the energy transition process implies the elaboration of public policies that in turn have to mirror the social, environmental and economic implications associated with the core concept of balance between energy consumption and production.

As such, I consider that the manuscript is relevant to the scope of SUSTAINABILITY.

To ensure it is fully fit for publication, please find my recommendations as follows:

1 Introduction – this section should guide more the readers through the relevant literature on the energy autarky concept and this aspect should be improved.
For example, after introducing the concept of autarky from an economic and political perspective, authors provide a reference list of relevant studies by others (see references 9-15, line 47). Nevertheless, authors should provide a short discussion of such bibliography, highlighting the scientific relevance of their argument and referring the lack/barriers on the knowledge, if any. Indeed, these aspects are core to make a clear presentation of the evidence-based background that justifies the two hypotheses proposed by the authors.

2 Methodology – I suggest authors refer to how they validate the results in comparison of other methods that exist in literature.

3 Results – I recommend the authors to review this section to make completely clear what is the originality of the model of energy self-sufficiency that they disclose.

4 The economic model of energy self-sufficiency that authors present in figure 2 is doubtful. I suggest to review it, focusing more on the parameters that are associated with renewable energy resources.

5 Conclusions are very simplistic and turn the paper less interesting to the journal readers. Please improve this section, including meaningful conclusions and concise recommendations.

In the end, I want to encourage the authors to continue their promising research on the concepts of energy self-sufficiency and energy self-reliance as a contribution toward a low carbon society transition.

Author Response

Thank you for your comments. All the comments have been taken into account. 

 1 Introduction – this section should guide more the readers through the relevant literature on the energy autarky concept and this aspect should be improved. 

The review of the relevant literature has been placed in the Introduction section.

2 Methodology – I suggest authors refer to how they validate the results in comparison of other methods that exist in literature.

Corrected.

 3 Results – I recommend the authors to review this section to make completely clear what is the originality of the model of energy self-sufficiency that they disclose.

4 The economic model of energy self-sufficiency that authors present in figure 2 is doubtful. I suggest to review it, focusing more on the parameters that are associated with renewable energy resources. 

Reviewed. The economic model (i.e. Figure 2) has also been reviewed.

5 Conclusions are very simplistic and turn the paper less interesting to the journal readers. Please improve this section, including meaningful conclusions and concise recommendations. 

Corrected.

Reviewer 2 Report

Geneal comments

The paper deals with the concepts of autarky and sustainability in the renewable energy local communities. The topic is of interest. Methodology is correct. Results are supported by data.

Specific comments

the English language Should be further checked. For example. Page 1, lines 38-44 the period is not clear. ".... achievable though ...." should be ".... achievable through ...."; page 9, line 386 when you speak about districts, please consider also:

Manos, B., Bartocci, P., Partalidou, M., Fantozzi, F., Arampatzis, S., Review of public-private partnerships in agro-energy districts in Southern Europe: The cases of Greece and Italy, (2014) Renewable and Sustainable Energy Reviews, 39, pp. 667-678

Fantozzi, F., Bartocci, P., D'Alessandro, B., Arampatzis, S., Manos, B., Public-private partnerships value in bioenergy projects: Economic, feasibility analysis based on two case studies, (2014) Biomass and Bioenergy, 66, pp. 387-397

Author Response

Thank you for your comments. All the comments have been taken into account.

the English language Should be further checked. For example. Page 1, lines 38-44 the period is not clear. ".... achievable though ...." should be ".... achievable through ...."; page 9,

Corrected.

line 386 when you speak about districts, please consider also:

Manos, B., Bartocci, P., Partalidou, M., Fantozzi, F., Arampatzis, S., Review of public-private partnerships in agro-energy districts in Southern Europe: The cases of Greece and Italy, (2014) Renewable and Sustainable Energy Reviews, 39, pp. 667-678

The example has been included in the Discussion section.

Reviewer 3 Report

This is a theoretical study, aimed at showing the role of the energy self-sufficiency (in terms of this paper - autarky) in the sustainable development concept. It also includes some information about the experience of German Bioenergy Villages and evolution of "bioenergy villages" concept.

In General, this article is an interesting piece of research. It is well-written, despite some problems with the structure. The topic and key conclusions are relevant, since many studies associate sustainable development with only transition to the renewables, without taking into account many other factors. Theoretical part seems rational and reasonable. The literature review is sufficient. I think that this article could be of interest to the readers.

Despite good General impression, I feel that this manuscript should be improved before acceptance. Here are my questions, comments and recommendations:

Line 47. References like [9-15] are not acceptable. Please, provide some kind of discussion for these articles. Should the footnote "1" be before the comma? Figure 1. I think it would be better to add some information to improve readability. But it is only my suggestions.

3.1. Targeted at relative autarky. If ">50%" goes to energy grid, should there be "<50%" lines to the households?

3.2. Absolute autarky. Please, add 100% label.

Line 120ff. The 50% threshold should have more strong justification. Figure 1. Please, add explanation to how we should use this model for a combination of energy carriers. For example, when power is required during the whole year and heat, which is provided fully/partly by external energy grid is required only in winter. Should we separate the carriers or we can make an integrated assessment? Methodology and results sections should be divided. Many theoretical information in section 3 relates to the literature analysis, which should be placed in Section 2. I feel that experience of the Germany's Villages should be showed in Results section, as a possible practical implication of energy autarky in terms of sustainable development strategy. It is also recommended to add statistical data about villages and their distribution, according to the proposed model of energy self-sufficiency. Maybe it will also be possible to show other indicators of their sustainable development. Discussion section should include overview of the findings and state their place in the existing knowledge, not only in Germany's experience. Please, add recommendations for further research in this area in Conclusion section.

Author Response

Thank you for your comments. Most of the comments have been taken into account.

Line 47. References like [9-15] are not acceptable. Please, provide some kind of discussion for these articles.

The review of the relevant literature has been placed in the Introduction section.

Should the footnote "1" be before the comma?

Corrected.

Figure 1. I think it would be better to add some information to improve readability. But it is only my suggestions. 

3.1. Targeted at relative autarky. If ">50%" goes to energy grid, should there be "<50%" lines to the households?

The energy to households in relative autarky model is exclusively provided by the external energy grid. There are no direct energy lines from local grids to households as the external energy grid mediates the transfers.

3.2. Absolute autarky. Please, add 100% label.

Both cases, i.e. absolute and relative autarky systems denote 100%. Autarky also assumes exceeded production (the energy can be sold). I clarified it in the text and the figure.

Line 120ff. The 50% threshold should have more strong justification.

Corrected.

Figure 1. Please, add explanation to how we should use this model for a combination of energy carriers. For example, when power is required during the whole year and heat, which is provided fully/partly by external energy grid is required only in winter. Should we separate the carriers or we can make an integrated assessment?

The autarky concept assumes the supply of 100 per cent of energy from local resources in a given period of time. If the time is defined as 1 year the total energy supply should meet the criterion. Thus, at the same time, we can say that a system is autarkical only in the terms of electricity (if the heating demand is not met in the analysed period, i.e. usually 1 year). It is explained in the following sentences: “Consequently, a local electricity system can be self-sufficient while a heating system is not. Moreover, the supply and demand for energy are asynchronous, and relative self-sufficiency in particular needs a time horizon [22,26]”. 

Methodology and results sections should be divided.

Done.

Many theoretical information in section 3 relates to the literature analysis, which should be placed in Section 2.

The review paragraphs have been changed.

I feel that experience of the Germany's Villages should be showed in Results section, as a possible practical implication of energy autarky in terms of sustainable development strategy. It is also recommended to add statistical data about villages and their distribution, according to the proposed model of energy self-sufficiency. Maybe it will also be possible to show other indicators of their sustainable development. Discussion section should include overview of the findings and state their place in the existing knowledge, not only in Germany's experience.

Germany is a good example of the model analysis as the idea was widely developed and research in this country. It is also in line with the topic of the paper. It illustrates the evolution due to the postulates of sustainability presented in this paper. Nevertheless, I also added some references to show other examples from Europe.  

Please, add recommendations for further research in this area in Conclusion section. 

Added.

Round 2

Reviewer 3 Report

Article could be published in present form.